# Theoretical investigation of a genetic switch for metabolic adaptation

Kathrin S. Laxhuber[1], Muir J. Morrison[2], Griffin Chure[3], Nathan M. Belliveau[4], Charlotte Strandkvist[5], Kyle L. Naughton[6], Rob Phillips[2,3]*

1 Department of Chemistry and Applied Biosciences, ETH Zurich, Zurich, Switzerland, 2 Department of Physics, California Institute of Technology, Pasadena, CA, United States of America, 3 Division of Biology and Biological Engineering, California Institute of Technology, Pasadena, CA, United States of America, 4 Howard Hughes Medical Institute, University of Washington, Seattle, WA, United States of America, 5 Department of Systems Biology, Harvard Medical School, Boston, MA, United States of America, 6 Department of Physics and Astronomy, University of Southern California, Los Angeles, CA, United States of America

* phillips@pboc.caltech.edu

**Data Availability Statement:** All simulation input and output files, code, and experimental data are available from DOI: 10.5281/zenodo.3733595.

## Abstract

Membrane transporters carry key metabolites across the cell membrane and, from a resource standpoint, are hypothesized to be produced when necessary. The expression of membrane transporters in metabolic pathways is often upregulated by the transporter substrate. In *E. coli*, such systems include for example the *lacY*, *araFGH*, and *xylFGH* genes, which encode for lactose, arabinose, and xylose transporters, respectively. As a case study of a minimal system, we build a generalizable physical model of the *xapABR* genetic circuit, which features a regulatory feedback loop via membrane transport (positive feedback) and enzymatic degradation (negative feedback) of an inducer. Dynamical systems analysis and stochastic simulations show that the membrane transport makes the model system bistable in certain parameter regimes. Thus, it serves as a genetic "on-off" switch, enabling the cell to only produce a set of metabolic enzymes when the corresponding metabolite is present in large amounts. We find that the negative feedback from the degradation enzyme does not significantly disturb the positive feedback from the membrane transporter. We investigate hysteresis in the switching and discuss the role of cooperativity and multiple binding sites in the model circuit. Fundamentally, this work explores how a stable genetic switch for a set of enzymes is obtained from transcriptional auto-activation of a membrane transporter through its substrate.

## Introduction

Genetic regulatory circuits are fundamental building blocks of functioning cells and organisms. One abundant class of these circuits are genetic switches. Although their construction and function may differ, their common feature is bistability: their output gene expression will flow to and remain at one of two steady-state levels. The distribution of gene expression in a cell culture can then be bimodal. This is not to be confused with mere stochastic bimodality, where the system is not stable, and the gene expression in each cell can fluctuate between the two levels.

**Funding:** This work was supported by the National Institutes of Health (https://www.nih.gov/) through 1R35 GM118043-01 Maximizing Investigators' Research Award (MIRA) (to R.P.), and the Werner Siemens Foundation (https://www.wernersiemens-stiftung.ch/en/) through the Swiss Study Foundation (https://www.studyfoundation.ch/) (K. S.L.). This material is based upon work supported by the National Science Foundation Graduate Research Fellowship (https://www.nsfgrfp.org/) under Grant No. DGE-1745301 (M.J.M.). The funders had no role in study design, data collection and analysis, decision to publish, or preparation of the manuscript.

**Competing interests:** The authors have declared that no competing interests exist.

One classic example of a genetic switch is a system where two repressor proteins each regulate the transcription of the other [1, 2] (illustrated schematically in Fig 1). Here, one stable state is high expression of the first protein and low expression of the second, and the second stable state is the opposite. This switch enables the system to have a memory: if something induces expression of either one of the proteins, the system will remain in this state until a significant perturbation occurs. Another well-known and even simpler example is an auto-activating circuit in which a protein activates its own transcription [3]. This gives the system an "on-off" switch.

Through physical and mathematical modeling, we investigate a more complex switch system where the bistability is due, as we will show, to a membrane transport protein. Such a switch is common for metabolic processes in biology, for reasons discussed below. Existing models in the literature tend towards one of two extremes: either highly detailed descriptions of specific, complicated networks (e.g., [4]), or Hill function descriptions that coarse-grain all complexity into a few parameters with inscrutable microscopic physical meaning. We aim for a middle ground in this work. We seek an intuitive understanding through a simple model of a minimal system, with only the essential components and interactions for the questions we pose. Yet we still model these components explicitly and discuss the necessary model complexity for a physically correct model.

The key feature of the type of system we investigate is the indirect activation of the transporter gene by the transporter substrate, leading to positive feedback similar to the aforementioned "on-off" switch. An example for such an architecture is the *lac* operon, where lactose indirectly activates the expression of lactose permease. Other examples in *E. coli* include the *araFGH* and *xylFGH* operons, which contain genes for arabinose and xylose transporters, respectively. For *lac* and *araFGH*, bistability has indeed been observed and attributed to such a positive feedback loop, for example in the well-known study by Novick and Weiner (1957), among other works [5–12]. A eukaryotic example is the glucose transporter GLUT-2 in liver and *β*-cells [13, 14], though this system is much more complex than the following analysis.

It is quite conceivable that this auto-activation process is common to many substances that a cell would want to consume. Such a switch enables the cell to sense and respond to its environment: if the substrate enters the cell, it activates the production of membrane transporters. The cell then starts accumulating the substrate, thereby "testing" the substrate's presence in the extracellular environment. If there is enough, the expression stabilizes at an "on" state and the cell has, in a short-term sense, adapted. When there is not enough substrate, the operon, which often encodes for a whole set of enzymes for this one metabolite, switches "off" again. Such a

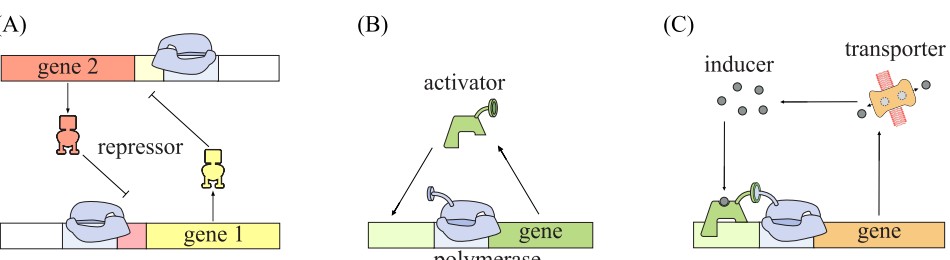

**Fig 1. A schematic of different genetic switches.** (A) and (B) show the two most well-known genetic switches: (A) two mutual repressors and (B) a self-activating gene. In (C), a very much simplified version of the circuit that we investigate in this paper can be seen, where the similarity to the switch in (B) is clear. A complete version of the model circuit can be found in Fig 3.

mechanism could plausibly be involved in various cases of short-term adaptation such as, but not limited to, the *lacZYA*, *araFGH*, and *xylFGH* examples mentioned above.

A key element of this mechanism is the presence of a transcription factor which binds to the transporter substrate and which is often expressed at a low level (often at copy numbers of order $\sim$ 10, [16, 27]). This is resource efficient for the cell, as this low copy number transcription factor acts as an "always on" sensor to detect the substrate, allowing high copy numbers of the membrane transporter and its attendant operon to be expressed only when their substrate is actually present. The transcription factors LacI, AraC, and XylR all appear to fill this role [5–12, 15, 16].

For our modeling, we focus on the *xapABR* genetic circuit from *E. coli* as a case study. It is similar to *lac*, but less complex. Instead of lactose, its purpose is to make use of the nucleoside xanthosine as an energy source [17, 18]. The circuit is made up of two operons: one that encodes for XapR and another that encodes for XapA and XapB. XapR is a transcription factor that is induced by xanthosine and activates the *xapAB* promoter, in close analogy to AraC, XylR, and also LacI. (One might object that LacI represses its target operon, while XapR, AraC, and XylR activate their target operons. However, the analogy we wish to draw is that the qualitative logic of their inducers are all identical, i.e., the presence of their respective inducer causes their target operon to be transcribed.) The *xapAB* promoter has been suggested to have two binding sites for XapR [19], but the promoter architecture and function is not yet fully understood. The transcription of *xapR* seems to be constitutive and not auto-regulated [19]. Structural homology to other transcription factors suggests that XapR appears in dimers where one dimer can bind two xanthosine molecules [20]. The protein XapA is a purine nucleoside phosphorylase that degrades xanthosine into components (ribose and xanthine) that can be fed into metabolic pathways [17, 18]. XapB on the other hand is a membrane transporter of xanthosine [19, 21].

Experimentally, we found that the expression level of *xapAB* among cells is bimodal and that the system seems to be bistable (see next section). We aim to understand which of the circuit's features are necessary for bistability and investigate its behavior in different parameter regimes. After presenting some experimental background on the *xap* genes, we discuss the details of our model. Lastly, we estimate the free parameters and then present the observations we made through phase diagrams, followed by the results from stochastic simulations.

## Experimental motivation

Our work was motivated by the experimental observation of bimodality in the *xap* circuit, which is shown in Fig 2. We focus on the essential findings here and refer the reader to S1 Text for more experimental details. Briefly, we placed a fluorescent reporter under the control of the wild type (wt) *xapAB* promoter. This construct was placed in three different backgrounds: Δ*xapABR*, Δ*xapAB*, and wt, and expression as a function of extracellular xanthosine concentration was measured using flow cytometry, as has been described previously [23]. The left panel of Fig 2A shows that for increasing xanthosine concentration, expression in the wt-background increases in a switch-like way: it is nearly zero when there is little xanthosine ("uninduced state"), but increases drastically when there is more ("induced state"). In between, the aforementioned bimodal distribution is obtained, where some bacteria are in the uninduced expression state and others are induced.

When all genes of the *xapABR* circuit are removed, the xanthosine response of our reporter construct disappears (see Fig 2A, middle panel). This is the result of removing the transcription factor XapR that is induced by xanthosine. When only the genes *xapAB* are removed but *xapR* is kept, a response to the xanthosine concentration is regained but it is no longer switch-

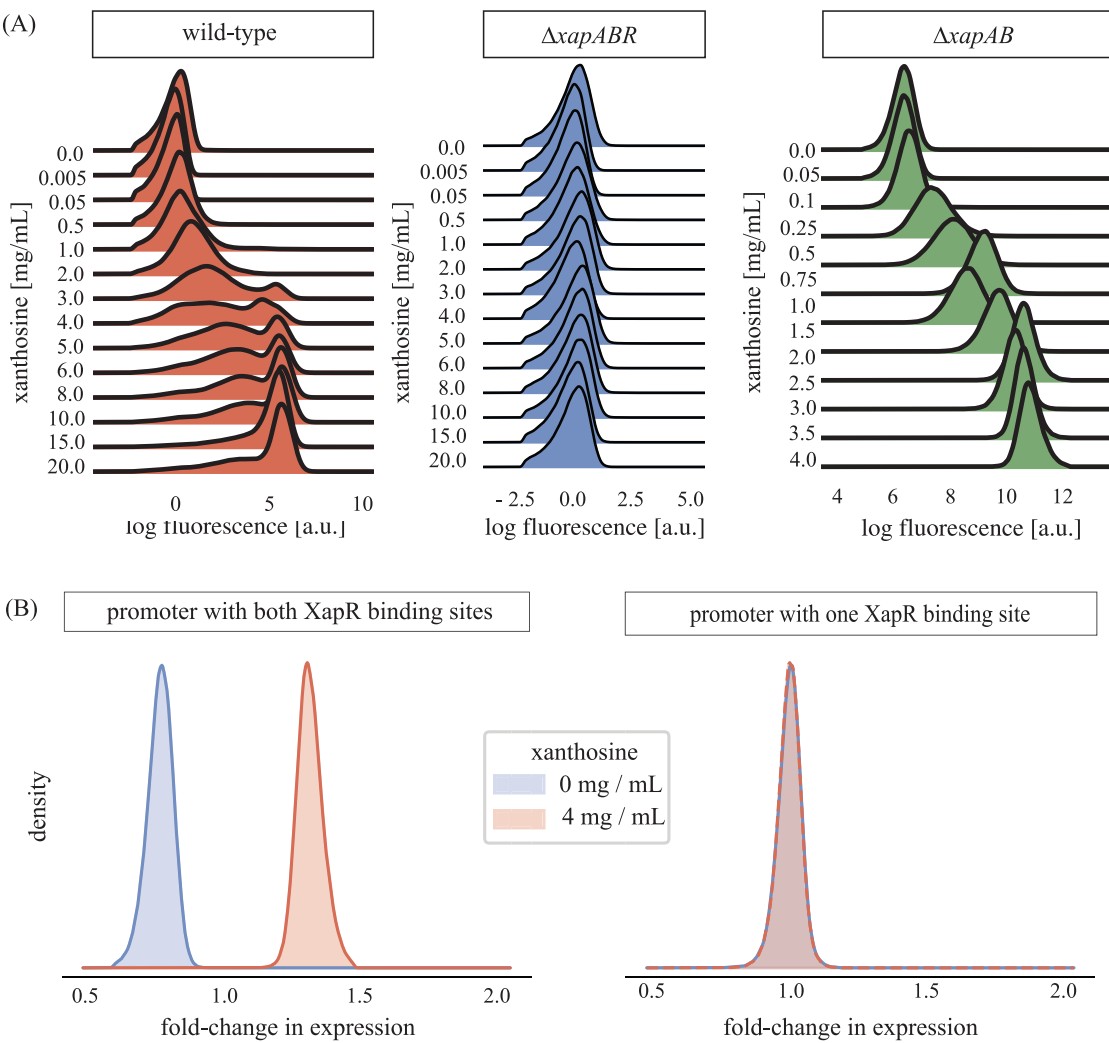

**Fig 2. Experimental data on the *xap* circuit.** (A) The expression of the *xapAB* promoter was measured for different extracellular concentrations of xanthosine (vertical axis). The left panel shows the wild type circuit while the middle and right panels show the effect of deleting the genes *xapABR* and *xapAB*, respectively. The wild type circuit behaves like a switch. Note that the fluorescence scale of the middle panel is not comparable with the other two, and also that the chosen xanthosine concentrations are different. (B) shows the fold-change in gene expression upon addition of xanthosine for the wt promoter (left panel) and for the promoter with only the XapR binding site adjacent to the polymerase binding site (right panel). Note that the fold-change used here differs from fold-change in, e.g., [22–24], in that no subtraction of autofluorescence was performed, which is adequate for the qualitative comparison of these two promoters.

like (see Fig 2A, right panel). Instead, the distribution remains unimodal and simply shifts to higher expression as the xanthosine concentration is increased. As we will see later on, this is because the circuit now lacks the positive feedback loop due to the xanthosine membrane transporter XapB.

As mentioned in the introduction, there are two binding sites for XapR on the *xapAB* promoter: one partially overlapping the polymerase -35 site and one further upstream. Working in the Δ*xapAB* background, we measured the expression level of our reporter when driven by two constructs: the wild-type *xapAB* promoter (Fig 2B, left panel), and the *xapAB* promoter with the upstream binding site removed (Fig 2B, right panel). Clearly, removing one XapR binding site dramatically reduced the responsiveness of the promoter to xanthosine (though

not apparent from Fig 2B, this response remains detectable in other measurements covering a smaller dynamic range).

Already, we believe these data, combined with the prior knowledge of the field, clearly suggest a minimal model of the *xapABR* system. The construction and exploration of this model occupies the remainder of this work. We leave it for future work to quantitatively dissect and test the model, in the manner of [22–24], for example.

## Model

### Step by step modeling of the system

In this section, we present our model of the *xapABR* genetic circuit. Fig 3 shows an overview of this model. The qualitative picture of the circuit switching its state is as follows:

- In the initial absense of XapB, small amounts of xanthosine permeate into and out of the cell (discussed in more detail below).

- The presence of xanthosine shifts XapR's equilibrium from inactive to active.

- Active XapR binds to the *xapAB* promoter, increasing transcription.

- From this mRNA transcript, translation produces the two proteins XapA and XapB.

- XapB actively transports much larger amounts of xanthosine into the cell and XapA degrades xanthosine.

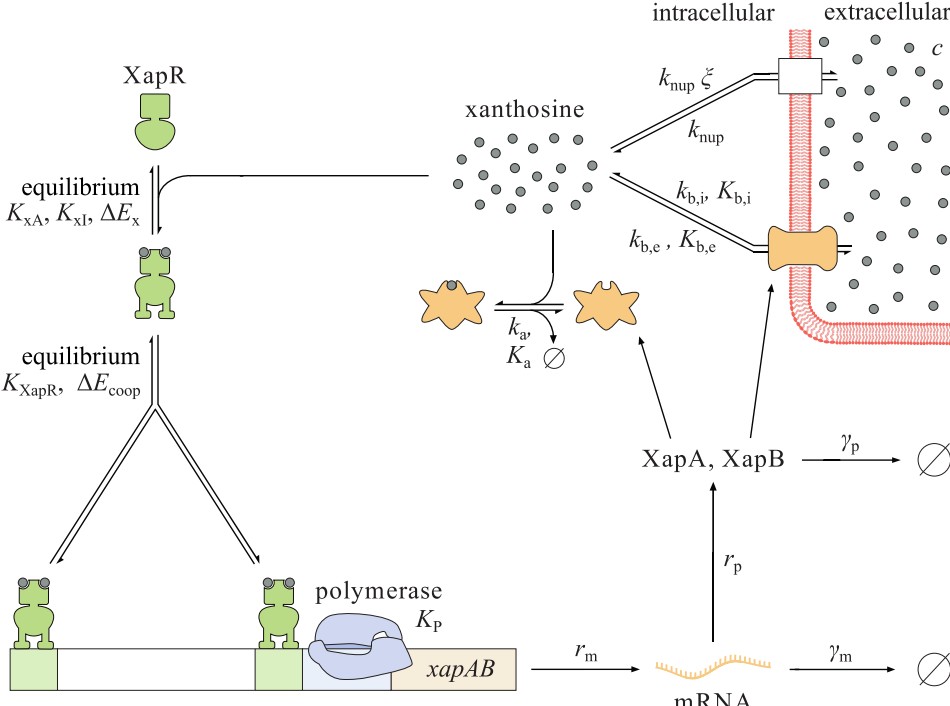

**Fig 3. Model of the *xapAB* circuit.** The XapR dimers are induced by xanthosine and the induced XapR binds cooperatively as an activator to the *xapAB* promoter. For these two steps, quasi-equilibrium is assumed. If both XapR binding sites are occupied and the polymerase is bound, the gene is transcribed at rate $r_m$. The mRNA decays at rate $\gamma_m$, and both proteins are translated at rate $r_p$ and decay at rate $\gamma_p$. XapA degrades xanthosine with Michaelis-Menten parameters $k_a$ and $k_a$. Similarly, XapB is treated as a Michaelis-Menten enzyme which imports ($k_{b,i}$, $K_{b,i}$) and exports ($k_{b,e}$, $K_{b,e}$) xanthosine. Furthermore, xanthosine enters and leaves the cell through non-specific transport, proportional to rates $k_{nup}$ and $\xi k_{nup}$, respectively.

- Production of more XapA and XapB is balanced by degradation or dilution through cell divisions.

Because xanthosine activates the transcription factor XapR, we have positive and negative feedback loops due to XapB and XapA, respectively. The remainder of this subsection discusses each of the above steps in detail, leading us to a set of two coupled ODEs. More in-depth explanations can be found in S1 Text.

**Induction of XapR.**   We treat dimers as the only form of XapR that appears in the cell. Each dimer can bind two xanthosine molecules [20]. The Monod-Wyman-Changeux (MWC) model is used to describe the fraction of XapR dimers in the active state, which has the form

$$[\text{XapR}]_\text{A} = [\text{XapR}]_\text{tot} \frac{\left(1 + \frac{[\text{x}]}{K_\text{xA}}\right)^2}{\left(1 + \frac{[\text{x}]}{K_\text{xA}}\right)^2 + e^{\beta \Delta E_\text{x}} \left(1 + \frac{[\text{x}]}{K_\text{xA}} \frac{K_\text{xA}}{K_\text{xI}}\right)^2}. \tag{1}$$

A detailed discussion of the MWC model can be found in [26]. In Eq 1, [x] is the xanthosine concentration, and $[\text{XapR}]_\text{A}$ and $[\text{XapR}]_\text{tot}$ denote the concentration of active and total XapR dimers, respectively. Furthermore, $K_\text{xI}$ and $K_\text{xA}$ are the dissociation constants of xanthosine to the inactive and the active XapR dimer, respectively, and $\Delta E_\text{x}$ stands for the energy difference between the inactive and the active states of the protein. We expect $\Delta E_\text{x} > 0$ and $K_\text{xA} < K_\text{xI}$ for inducible activation. This corresponds to XapR being mainly inactive in the absence of xanthosine and becoming mostly active at high concentrations of xanthosine.

**Transcription.**   Transcription and translation of the *xapAB* operon, regulated by the induced XapR, produce the two proteins XapA and XapB. We start with transcription and assume that the binding of XapR and polymerase to the promoter is at quasi-equilibrium. The polymerase binding is modeled as independent of that of XapR, and all influence of the activator is pushed into the transcription rate. Furthermore, the binding energy of XapR to each of its two sites is assumed to be the same. A discussion of these simplifications can be found in S1 Text.

In Fig 4, all possible states of the promoter in our model and their corresponding thermodynamic weights are shown. [P] denotes the polymerase concentration, and $\Delta E_\text{coop}$ stands for the interaction energy of the two XapR dimers. If cooperativity in transcription factor binding is neglected, this is set to zero. Furthermore, $K_\text{XapR}$ and $K_\text{P}$ denote the dissociation constant of XapR and polymerase to the promoter, respectively. In statistical mechanics language these dissociation constants are equivalent to $\frac{N_\text{NS}}{V} e^{\beta \Delta E_\text{XapR}}$ and $\frac{N_\text{NS}}{V} e^{\beta \Delta E_\text{P}}$, respectively, with $N_\text{NS}$ being the number of non-specific binding sites on the DNA, $V$ the volume of the cell, and $\Delta E_\text{XapR}$ and $\Delta E_\text{XapR}$, respectively, the interaction energies of XapR or polymerase with the promoter.

We consider only the state where both XapR binding sites are occupied as active and all other states as inactive, meaning they have transcription rate equal to zero. Our experiments show that the expression becomes very weak when one of the XapR binding sites is removed from the promoter (see Fig 2B and S1 Text), suggesting that this simplification is reasonable. Furthermore, we find that in the bistable parameter range, considering the single occupancy states as active instead has almost no influence on the results (see also S1 Text).

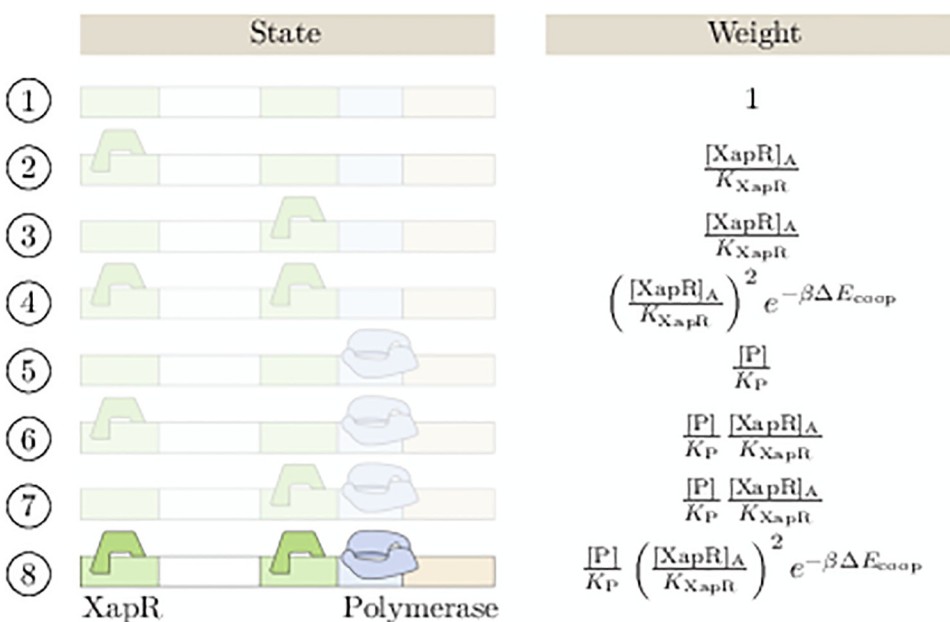

**Fig 4. The promoter states.** We consider only the completely occupied state as active and all other states (faded out in the figure) as completely inactive. The parameters are the interaction energy of the two XapR dimers $\Delta E_{\mathrm{coop}}$ and the dissociation constants $K_{\mathrm{XapR}}$ and $K_{\mathrm{P}}$ of XapR and polymerase to the promoter, respectively. The concentrations of polymerase and active XapR are denoted by [P] and [XapR].

With [m] being the mRNA concentration, $r_{\mathrm{m}}$ the transcription rate, $\gamma_{\mathrm{m}}$ the mRNA decay rate, and $p_{\mathrm{active}}$ the probability of the promoter being in the active state, we obtain

$$\frac{\mathrm{d}[m]}{\mathrm{d}t} = r_{\mathrm{m}} p_{\mathrm{active}} - \gamma_{\mathrm{m}}[m] \tag{2}$$

$$p_{\mathrm{active}} = \frac{w_8}{\sum_{i=1}^{8} w_i} = \frac{[P]}{K_{\mathrm{P}} + [P]} \frac{\left(\frac{[\mathrm{XapR}]_A}{K_{\mathrm{XapR}}}\right)^2 e^{-\beta \Delta E_{\mathrm{coop}}}}{1 + 2\frac{[\mathrm{XapR}]_A}{K_{\mathrm{XapR}}} + \left(\frac{[\mathrm{XapR}]_A}{K_{\mathrm{XapR}}}\right)^2 e^{-\beta \Delta E_{\mathrm{coop}}}} \tag{3}$$

Here, $w_i$ stands for the thermodynamic weight of the ith state in the order in which they are listed in Fig 4. As written above, the partition function factorizes into a polymerase and a XapR term because of our assumption of independent binding, which is further discussed in S1 Text. Note that because $r_{\mathrm{m}}$ implicitly contains the gene copy number per cell, it has units of $\mathrm{M}^{-1}\,\mathrm{s}^{-1}$ and not just $\mathrm{s}^{-1}$. This rate equation gives the mean mRNA concentration $\langle[\mathrm{mRNA}]\rangle = \frac{r_{\mathrm{m}}}{\gamma_{\mathrm{m}}} p_{\mathrm{active}}$, which we will need in the next paragraph. The mean can also be found from the full chemical master equation, which is provided in S1 Text.

**Translation.** The next step in our modeling progression is translation. As a simplification, we write [p] = [XapA] = [XapB] for the general protein concentration. This assumes that the rates of transcription, mRNA decay, translation, and protein decay are the same for both proteins, which, as discussed in S1 Text, does not have a significant influence on the results. We write the following rate equation for the protein concentration, where $r_{\mathrm{p}}$ denotes the

translation rate, $\gamma_p$ the protein decay rate, and $\langle[m]\rangle$ the mean mRNA concentration:

$$\frac{d[p]}{dt} = r_p\langle[m]\rangle - \gamma_p[p].$$
(4)

**Xanthosine dynamics.**    Having described how xanthosine activates the synthesis of XapA and XapB through XapR, we now close the feedback loop by setting up a xanthosine rate equation.

There are two significant mechanisms for transport of xanthosine across the cell membrane. In the induced system, the main transporter is XapB, whereas in the uninduced system, there is almost no XapB. Instead, xanthosine can enter the cell through the two nucleoside transporters NupC and NupG, which have a very low affinity for xanthosine [21]. All these transporters, XapB, NupC, and NupG, are powered by the proton gradient across the membrane [21], which is why we assume their kinetic scheme to be similar to that of the *lac* permease (as it is described in [26]). There can be import and export of xanthosine, and which one dominates depends on the proton and xanthosine concentrations on the two sides of the membrane. In both cases, a proton and a substrate need to bind to the transporter on one side of the membrane and detach from it on the other side before the empty transporter moves back to the other side again. We refer the reader to S1 Text for a detailed description of the transport and its modeling, and just state the result here. We model influx and efflux separately. For XapB, we use Michaelis-Menten kinetics with parameters $k_{b,i}$, $K_{b,i}$ for influx and $k_{b,e}$, $K_{b,e}$ for efflux. For the Nup transporters, we also use Michaelis-Menten kinetics but, because of the transporter's low affinity for xanthosine, we can linearize the Michaelis-Menten term as $k_{cat}[x]/(K_M + [x]) \approx k\,[x]$ (i.e., $K_M \gg [x]$ across the physiologically relevant range for [x]). For the rate parameters $k$, we write $k_{nup}$ for influx and $\xi k_{nup}$ for efflux.

After transport into the cell, XapA degrades xanthosine. We model this using standard Michaelis-Menten kinetics, with parameters $k_a$, $K_a$ (corresponding to turnover rate and Michaelis constant, respectively). Transport and degradation then leads to the xanthosine rate equation

$$\frac{d[x]}{dt} = \left( \underbrace{k_{b,i}\frac{c}{K_{b,i}+c} - k_{b,e}\frac{[x]}{K_{b,e}+[x]}}_{\text{XapB}} - \underbrace{k_a\frac{[x]}{K_a+[x]}}_{\text{XapA}} \right)[p] + \underbrace{k_{nup}(c - \xi[x])}_{\text{NupC \& NupG}}.$$
(5)

Recall that [x] is the intracellular xanthosine concentration, while $c$ denotes the extracellular concentration. Because $k_{b,i} > k_{b,e}$ and $K_{b,i} < K_{b,e}$, influx dominates at low intracellular xanthosine concentrations. At much higher intra- than extracellular xanthosine concentrations, the efflux term takes over. More details on the aforementioned steps and a discussion of passive diffusion can be found in S1 Text.

## Nondimensionalization

We have now formulated the behavior of the system in terms of the rate equations for mRNA, protein, and xanthosine. These equations can be nondimensionalized, which reduces the dimension of parameter space. We measure time in units of $\gamma_p^{-1}$ and concentrations in units of $k_a$ (except XapR, where the equations make it more natural to use $K_{XapR}$). In Table 1, all the nondimensional parameters and their definitions are listed. Furthermore, we define $[m]_a := \frac{[m]}{K_a}$, $[p]_a := \frac{[p]}{K_a}$, $[x]_a := \frac{[x]}{K_a}$, $[c]_a := \frac{[c]}{K_a}$, and $\tau := \gamma_p t$. Using these definitions, the following equations are

**Table 1. Nondimensional parameters and their estimated values.**

| Param. | Definition | Estimated range | Value used |
|---|---|---|---|
| $\rho_m$ | $:= \frac{r_m}{\gamma_p K_a} \frac{[P]}{K_P + [P]}$ | $\approx 10^{-3 \pm 2}$ | $10^{-3}$ |
| $\gamma_{mp}$ | $:= \frac{\gamma_m}{\gamma_p}$ | $\approx 10^{1 \pm 0.5}$ | $10^1$ |
| $\rho_p$ | $:= \frac{r_p}{\gamma_p}$ | $\approx 10^{2 \pm 0.5}$ | $10^2$ |
| $[XapR]_R$ | $\frac{[XapR]_{tot}}{K_{XapR}}$ | $\approx 10^{0 \pm 2}$ | $10^0$ |
| $[c]_a$ | $\frac{c}{K_a}$ | $(\in [0, 10^3])$ | 13 |
| $k_{\beta,i}$ | $:= \frac{k_{b,i}}{\gamma_p}$ | $\approx 10^{4 \pm 1}$ | $5 \cdot 10^4$ |
| $k_{\beta,e}$ | $:= \frac{k_{b,e}}{\gamma_p}$ | $\approx 10^{3 \pm 2}$ | $10^3$ |
| $k_\alpha$ | $:= \frac{k_a}{\gamma_p}$ | $\approx 10^{2 \pm 0.8}$ | $10^2$ |
| $k_\eta$ | $:= \frac{k_{mp}}{\gamma_p}$ | $\approx 10^{0 \pm 3}$ | $5 \cdot 10^{-1}$ |
| $\xi$ | $= \xi$ | $\approx 0.8 \pm 0.1$ | 0.8 |
| $K_{\beta,i}$ | $:= \frac{K_{b,i}}{K_a}$ | $\approx 10^{1 \pm 2}$ | $10^1$ |
| $K_{\beta,e}$ | $:= \frac{K_{b,e}}{K_a}$ | $\approx 10^{2 \pm 2}$ | $10^2$ |
| $K_{\chi A}$ | $:= \frac{K_{xA}}{K_a}$ | $\approx 10^{2 \pm 1} \cdot 10^{\Delta\epsilon_x - 5}$ | $10^2$ |
| $K_{IA}$ | $:= \frac{K_{xI}}{K_{xA}}$ | $\approx 10^{2 \pm 1}$ | $10^2$ |
| $\Delta\epsilon_x$ | $:= \beta\Delta E_x$ | $\approx 2$ to $2(\ln(K_{IA}) - 1) < 12$ | 5 |
| $\Delta\epsilon_{coop}$ | $:= \beta\Delta E_{coop}$ | $\approx 0 - 10$ | 5 |

The left column shows all nondimensional parameters that appear in the final equations. In the middle are their definition and estimated values. They are based on $\gamma_p = 5 \cdot 10^{-4}\ \text{s}^{-1}$ and $K_a = 5 \cdot 10^{-5}$ M. Note that the range of the three MWC parameters depends on each other, but they can still be chosen independently. The range given for $[c]_a$ denotes the estimated "interesting" range in which switching happens, but $[c]_a$ can of course exceed these values. Details on the parameters and their estimation can be found in S1 Text. Finally, the last column shows the value that we use for the rest of this paper, unless otherwise noted. An explanation of this choice will follow in the next section.

obtained

$$\frac{d[m]_a}{d\tau} = \rho_m \frac{[XapR]_{R,A}^2 e^{-\Delta\epsilon_{coop}}}{1 + 2[XapR]_{R,A} + [XapR]_{R,A}^2 e^{-\Delta\epsilon_{coop}}} - \gamma_{mp}[m]_a \tag{6}$$

$$\frac{d[p]_a}{d\tau} = \rho_p[m]_a - [p]_a \tag{7}$$

$$\frac{d[x]_a}{d\tau} = \left( k_{\beta,i} \frac{[c]_a}{K_{\beta,i} + [c]_a} - k_{\beta,e} \frac{[x]_a}{K_{\beta,e} + [x]_a} - k_\alpha \frac{[x]_a}{1 + [x]_a} \right)[p]_a$$
$$+ k_\eta([c]_a - \xi[x]_a) \tag{8}$$

$$\text{with} \quad [XapR]_{R,A} = [XapR]_R \frac{\left(1 + \frac{[x]_a}{K_{\chi A}}\right)^2}{\left(1 + \frac{[x]_a}{K_{\chi A}}\right)^2 + e^{\Delta\epsilon_x}\left(1 + \frac{[x]_a}{K_{\chi A}}\frac{1}{K_{IA}}\right)^2}$$

Very little is known about the *xap* system, and thus, there are almost no measured values for the free parameters. Nevertheless, we were able to estimate a reasonable range by using values from similar, well studied systems and by exploiting physical constraints or relations between parameters. The results of these estimates are shown in Table 1. They are based on a choice of $\gamma p = 5 \cdot 10^{-4}\, s^{-1}$ and $K_a = 5 \cdot 10^{-5}$M. A detailed derivation can be found in S1 Text.

## Results and discussion

In the modeling process in the previous section, we have obtained three coupled differential equations. In this section, we will analyze these equations with deterministic methods and stochastic simulations. Analytical closed-form solutions could not be obtained and would, if they existed, probably not be helpful due to their large complexity. Finding such solutions requires solving a fifth order algebraic equation.

### Deterministic phase portraits

A standard way to analyze dynamical systems deterministically is to plot phase portraits. In the following, we present such plots where the state variables are the mRNA, the protein, and the xanthosine concentration. Note that for the low copy numbers that can occur in our system, a deterministic analysis is not necessarily meaningful. Nevertheless, in our case we find that stochastic simulations agree well with the deterministic results. Thus, deterministic phase portraits are a valid starting point.

**From a 3D to a 2D system.** Fig 5A shows the 3D phase portrait for a representative set of parameters (shown in Table 1), whose choice is explained below. The plot looks rather complicated at first but can be understood intuitively. The three surfaces are the nullcline surfaces and the gray lines point in the direction in which the dynamical system moves at each point. The surfaces intersect in three points, which are the steady-state solutions of the dynamical system. For this choice of parameters, the system first flows towards the mRNA nullcline (independent of the initial condition), then it moves along that surface to the intersection with

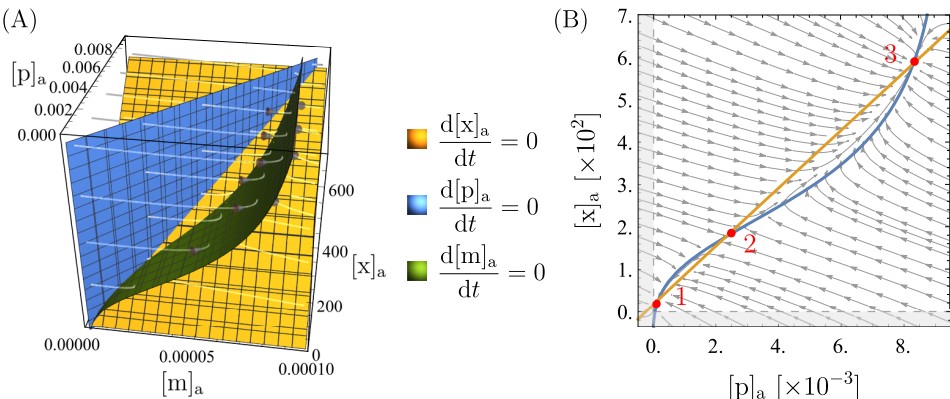

**Fig 5. Phase portraits showing bistability.** 3D and 2D phase portraits for one set of parameters that leads to bistability. The parameter values are listed in Table 1. Note that all the concentrations ($[m]_a$, $[p]_a$, $[x]_a$) are measured in units of $K_a = 5 \cdot 10^4$ nM. The surfaces in (A) and the curves in (B) are the nullclines of the state variables, and their intersection points, marked in red in (B), are the steady-state solutions of the system. The region shaded in gray in (B) leads to negative concentrations and is unphysical. A vector plot of (B) that also shows the magnitude of flow at each point can be found in S1 Text.

the protein nullcline, and lastly, it moves along that intersection line to one of the three intersection points of all three surfaces.

It is important to point out that, for a different set of parameters, the dynamics can be quite different. There are, for example, scenarios where the xanthosine kinetics are roughly as fast as the mRNA kinetics and the dynamics unfolds in two steps: first to the intersection of the mRNA and the xanthosine nullcline, then along that curve to the protein nullcline and thereby to a fixed point.

A usual simplification with genetic circuits like this is to assume the mRNA concentration to be at steady-state, i.e., to write $\frac{d[m]_a}{d\tau} = 0$ and solve this for $[m]_a([p]_a, [x]_a)$ to simplify the 3D to a 2D system. This restricts the dynamics to the green surface in our plot, which is reasonable here because as explained above, the system first flows towards that surface before either the protein or the xanthosine concentration changes significantly. However, as already pointed out, this is different for other parameter values, and thus, this assumption does not hold in general. If the xanthosine dynamics are faster than the mRNA dynamics, the system first flows towards the xanthosine nullcline. In that case, forcing it onto the mRNA nullcline leads to significant changes in the dynamics.

Nevertheless, the steady-state solutions and the qualitative features that we address in this paper remain the same. Because the 3D plots are rather hard to read, we will, in the following, make the compromise to show a 2D version of the phase portraits but ensure that all of our statements also hold true in 3D space. As explained above, it makes the most sense here to do this by setting $\frac{d[m]_a}{d\tau} = 0$. The resulting equations can be found in S1 Text. In particular, we define $\rho := \frac{\rho_m \rho_p}{\gamma_{mp}}$ for everything that follows.

**Bistability.** We map the mRNA nullcline surface (green in Fig 5A) onto a plane to show it as the 2D plot in Fig 5B. From this 2D plot, it can clearly be seen that for the chosen parameters, there are three steady-state solutions. Because the system is restricted to the mRNA nullcline surface, these steady-state solutions are the same as those in the 3D plot ($\frac{d[m]_a}{d\tau} = 0$ on the nullcline and $\frac{d[p]_a}{d\tau} = 0, \frac{d[x]_a}{d\tau} = 0$ for the 2D fixed points). One can see from the vector field that the two outer fixed points (labeled 1 and 3) are stable and the middle one (labeled 2) is unstable and serves as a sort of "switch-point" between the other two. This means that there are two stable states the cell can be in, one at high (point 3) and one at low (point 1) expression. As a result, there is bistability and the distribution of expression among cells can be bimodal, depending on initial conditions.

The bistability corresponds to the experimental observations, so the model passes this sanity check. Furthermore, the xanthosine and protein concentrations at the upper fixed point have the expected order of magnitude: the xanthosine concentration is roughly $10 - 100$ mM, and there are roughly 500 proteins, which is just a bit lower than what was measured for the number of Nup transporters [16] which fulfill a similar purpose. We do not have well founded expectations for the other fixed points, so no comparison can be made here. Nevertheless, the orders of magnitude at the lower fixed point—roughly $1 - 10$ nM of xanthosine and around 5 proteins—seem quite reasonable. Note that $[x]_a \approx [c]_a$ at the lower fixed point because there is only weak accumulation due to Nup and a few XapB transporters.

As already mentioned, we are working with one specific set of parameters here and we will now explain this choice of values. Firstly, they were picked roughly in the middle of the range that was estimated beforehand for this parameter (see Table 1 and S1 Text). Secondly, we chose parameters that allow clear bistability in the phase portraits as well as in the stochastic simulations (see later), which, of course, is not the case for any possible choice of parameters. Thirdly, by the corresponding choice of parameters it was ensured that the mRNA number per

cell at the "switch-point" is around 1: this is large enough to enable the system to clearly resolve the two stable fixed points (as we will see from the stochastic simulations later on), but is low enough to lead to mean mRNA numbers that are very reasonable (of order $10^{-2}$ to $10^1$ transcripts per gene per cell in *E. coli* [28]). The protein and xanthosine concentrations followed from this, but with some variation in the parameters they could still be tuned to a certain extent.

We point out that we have not observed any oscillations in the system. Intuitively, they might be expected when the XapA rate is significantly larger than the XapB rate, but it turns out that oscillations cannot be obtained. Why they do not occur can be understood when looking at the regions that are bounded by all three nullclines: on these boundaries, the streamlines point into the bounded regions, so deterministically, they serve as trapping regions from which the system cannot escape. Once inside, the only possible trajectory is non-oscillatory flow towards the stable fixed point.

For a different set of parameters, the orders of magnitude in the plots and even the qualitative behavior can change. In the following, we will discuss some interesting features of the system that can be observed through the phase portraits.

**The extracellular xanthosine concentration.** The parameter that is the experimentally most easily tunable and biologically the most relevant is the extracellular xanthosine concentration. When it is increased in experiments, the cells go from (1.) all being in the low expression state to (2.) the population being in a mixed state with some cells in a low expression state and others in a high expression state (all-or-none phenomenon) and then to (3.) all being in the high expression state (see section "experimental motivation"). If our model is correct, it should exhibit the same qualitative behavior. Indeed we find exactly this: as can be seen in Fig 6, increasing $[c]_a$ makes the high stable fixed point appear and then, for even higher $[c]_a$, the lower one disappears. Thus, for low $[c]_a$ the only stable point of the system is at low expression, and for high $[c]_a$ there is only high expression. In between, there are two deterministically

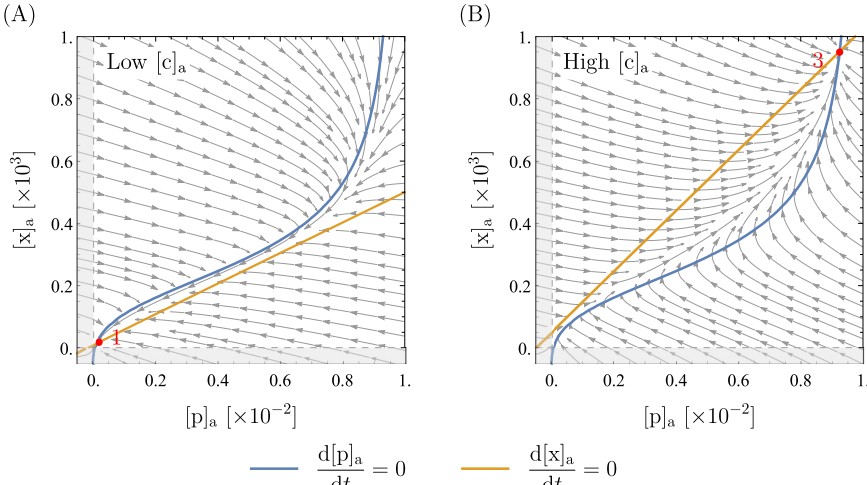

**Fig 6. Phase portraits for different extracellular xanthosine concentrations.** All parameters but $[c]_a$ are as presented in Table 1. The extracellular xanthosine concentration in these plots is $[c]_a = 7$ in (A) and $[c]_a = 40$ in (B) (recall that $[c]_a := \frac{c}{K_a}$ with $K_a = 5 \cdot 10^{-5}$M, so $[c]_a$ is dimensionless). Tuning $[c]_a$ moves the orange line (xanthosine nullcline), but the blue curve (mRNA nullcline) is unchanged (see also S1 Text). It can clearly be seen that in (A) there is only the lower fixed point (fixed point number 1), whereas in (B) there is only the upper one (fixed point number 3). In between lies the bistable case that was shown in Fig 5.

stable expression levels. Another means of visualizing this is with an bifurcation diagram, which we discuss in detail later.

Furthermore, we found that in the absence of xanthosine, i.e., setting $[c]_a = 0$(not shown here), there are roughly 2-3 copies of XapA and XapB, which agrees very well with measurements, where around 2 copies per cell were found [27]. In addition, the parameter $K_{\chi A}$ (dissociation constant of xanthosine from active XapR) can be tuned such that the extracellular xanthosine concentration $[c]_a$ in the switching-regime is similar to that in the experiment. It was found that the cell only adapts at very high xanthosine concentrations of almost a millimolar [21] which is not completely unexpected when recalling that for *lac*, cells also limit themselves to glucose as long as possible. Interestingly, because there is no parameter other than $K_{\chi A}$ that tunes the critical value of $[c]_a$, this tells us that $K_{\chi A}$ is large as argued in the estimation of $K_{\chi A}$ in S1 Text. Thus, we predict that the interaction between xanthosine and XapR should be weak.

**The roles of XapA and XapB.** While it is clear that the bistability in the model system is due to the feedback loop from XapA and XapB, it is not intuitively clear if both XapA and XapB are necessary. The model implies that the bistability is due to XapB only. XapA is neither sufficient nor necessary and, within the estimated parameter regime, does not even have a significant influence on the system. This can be seen from the plots in Fig 7. Degradation of xanthosine by XapA lowers the xanthosine and protein concentration at the upper fixed point by a small amount and could, in principle, thereby make the high-expression solution vanish. For our choice of all other parameters, bistability only vanishes for $k_\alpha > 10^4$ which is far from what has been measured. However, a higher effective rate could, in principle, be achieved by different translation rates of XapA and XapB (see simplifications of the model in S1 Text). Hence, we cannot exclude the possibility that XapA becomes so strong that it makes bistability impossible, but this is an extreme case. XapB, on the other hand, is essential; without it the system only has the one fixed point at low expression.

For a cell, the minimal effect of XapA on bistability is a useful feature: by coupling XapA and XapB on an operon, XapA is switched on and off together with XapB but it does not

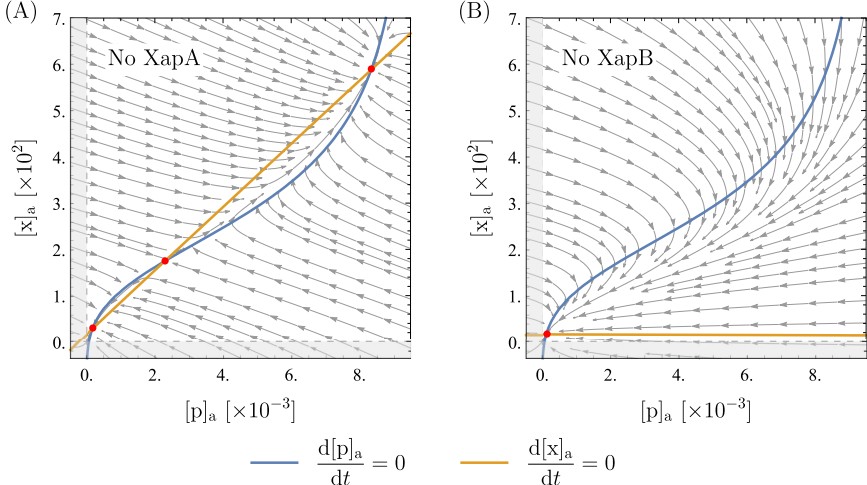

**Fig 7. Phase portraits without XapA or XapB.** All parameters are as presented in Table 1. In (A), the XapA term was removed from the kinetic equations. In (B), the equations lack the two terms from XapB. These plots clearly show that XapA has almost no influence on the qualitative behavior of the system (i.e. bistability and the order of magnitudes), but XapB is the essential feature for bistability.

significantly disturb this adaptation mechanism, while its kinetic parameters and expression levels can be chosen somewhat freely as necessary for metabolism. By having a membrane transporter gene on an operon whose expression is activated by the transporter substrate, the expression of a whole set of enzymes can be turned on and off depending on the presence of the substrate. It seems likely that this mechanism of short-term adaptation of a single cell to its environment may be used by cells for many metabolic processes.

As a side note, Seeger et. al. [19] observed that Δ*xapB* mutants could survive, but grew extremely slowly, with xanthosine as the only carbon source. This suggests that low-affinity import of xanthosine by NupC and NupG is sufficient to sustain slow growth, but insufficient to serve as a stand-in for XapB. Fig 7 supports this supposition: in our model, with *xapB* removed, the switch never activates and the cells are forced to survive with an extremely meager quantity of XapA to metabolize the abundant xanthosine.

**The role of cooperativity.** The model has two (putatively) cooperatively interacting binding sites for XapR on the *xapAB* promoter and two cooperative binding sites on XapR for xanthosine. It is interesting to consider whether the cooperativity is a necessary feature for bistability. This question is motivated by the importance of cooperativity in "typical" genetic switches [2, 29].

If, as a purely theoretical consideration, we remove either the second xanthosine binding site on XapR or the second XapR binding site on the promoter, leaving cooperativity in only one component of the system, we find that the system still has a bistable parameter regime. However, this bistable parameter range is smaller than in the original model, which makes the system less stable: small stochastic fluctuations in the parameter values can collapse the system to monostability, possibly leaving it in the wrong state and without its ability to adapt. But only when the second binding site is removed in both places, leaving no cooperativity in the system, do we find that it is insufficiently non-linear to produce bistability. An example of the three scenarios (only cooperative XapR, only cooperative promoter, no cooperativity) can be seen in Fig 8. It follows that there need to be either two xanthosine binding sites on XapR or two XapR binding sites on the promoter (or both) in order to obtain a switch-like behavior.

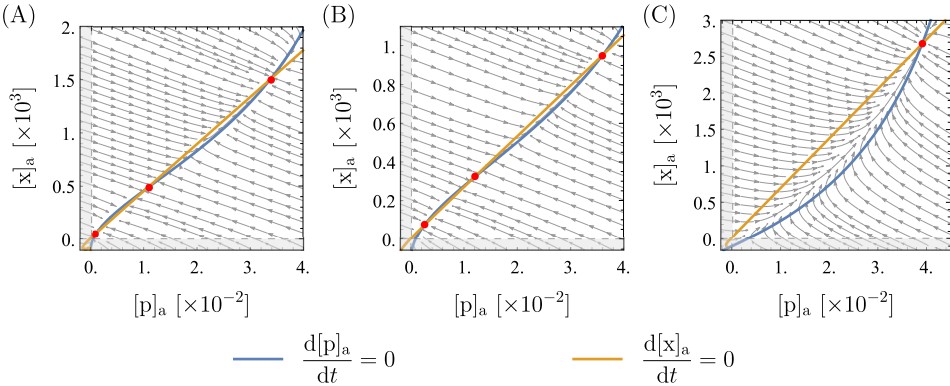

**Fig 8. Phase portraits for less or no cooperativity.** Most parameters are as presented in Table 1, changes are mentioned below. Fixed points are marked in red. In (A), there is only one xanthosine binding site on XapR and everything unchanged for the XapR-promoter binding. Two parameters are changed: $\rho = 0.07$ and $[c]_a = 6$. This is necessary to compensate for the weaker induction such that the system is bistable. In (B), there is only one XapR binding site on the promoter and everything is unchanged for the xanthosine-XapR binding. Two parameters are changed: $\rho = 0.13$ and $[c]_a = 3$. In (C), there is only one xanthosine binding site on XapR and also only one XapR binding site on the the promoter. Two parameters are changed: $\rho = 0.1$ and $[XapR]_R = 5$. Whereas bistability is retained in (A) and (B), it cannot be obtained anymore in (C).

One can also ask how much cooperative interaction is needed between the two binding sites. For the promoter, the amount of cooperativity is given by $\Delta E_{\mathrm{coop}}$ in our model, and we find that setting $\Delta E_{\mathrm{coop}} = 0$ has almost no influence on the phase diagrams. For XapR, we cannot test how much interaction is needed: the two binding sites interact indirectly, because the active state is much likelier if two xanthosine molecules are bound, and thus there is no continuous tuning parameter for the cooperative interaction like $\Delta E_{\mathrm{coop}}$ in the case of the promoter.

Note that we are not writing Hill equations and measuring cooperativity in terms of the Hill coefficient. If Hill equations were to be used for the modeling, the Hill coefficient could have values between 1 and 2, which would yield bistability for large enough values, but not for lower ones. This could be investigated more rigorously similar to the analysis of a simple genetic switch in [29]. However, we refrain from looking for a minimal Hill coefficient in our system, because we do not find this very insightful. Hill equations only describe some specific limit cases of cooperative systems, but for example do not account for interaction energies and assume the partially bound states (e.g. only one XapR bound to the promoter) to never be populated. We suggest that cooperativity should be explored more in-depth and a more rigorous analysis of the role of cooperativity in simple genetic switches should be done before returning to more complex systems like this one.

## Stochastic simulations

Stochastic simulations of the full 3-dimensional system of mRNA, protein, and xanthosine were run for comparison with the deterministic results. In S1 Text, we present the underlying chemical master equation of the system. Because of the two different fixed points at low and high expression, the protein copy numbers in the problem vary from less than five to several thousand. Even worse, xanthosine copy numbers may range as high as $10^7$ at the high expression fixed point. For such large copy numbers, the number of reaction firings that must be simulated with Gillespie's classical algorithm [30] leads to an impractical computational cost. This would make Gillespie's $\tau$-leap algorithm [31] ideal for the high expression state. On the other hand, $\tau$-leaping cannot be used for the small protein copy numbers in the low expression state, or the mRNA copy number which remains of order ten or less in both states. For these reasons, we chose to work with the algorithm described in [32], a hybrid form between Gillespie's classical and his $\tau$-leap algorithm. We gratefully worked with the Python implementation of this algorithm in *StochPy*, version 2.3 [33].

Note that we neglect stochastic fluctuations in any of our parameters, in particular in the XapR copy number. The latter is on the order of 10, but because of the long life span and rare expression of proteins like XapR, we expect the influence of the simplification on our results to be rather small. Nevertheless, the overall stochastic fluctuations are expected to be larger in the real system than in our simulations.

**Bimodality and the extracellular xanthosine concentration.** The stochastic approach results in the same bimodal distributions that were already seen in the deterministic investigation and experimental studies. Fig 9 shows the distribution of protein expression found in the simulations for different values of the extracellular xanthosine concentration. The parameters that were used are the same as in the previous section (listed in Table 1). However, we also find good agreement of the simulations and the deterministic results for other sets of parameters. To obtain the distributions, we ran the simulation 5000 times for a simulated time of $10^6$ s each and started at an mRNA, protein and intracellular xanthosine count of 0.

The results agree very well with the deterministic fixed points and experiments: the mean numbers of mRNA, protein, and xanthosine in the stochastic results are as predicted from the phase portraits. It does, however, become clear that the phase portraits do not tell whether the

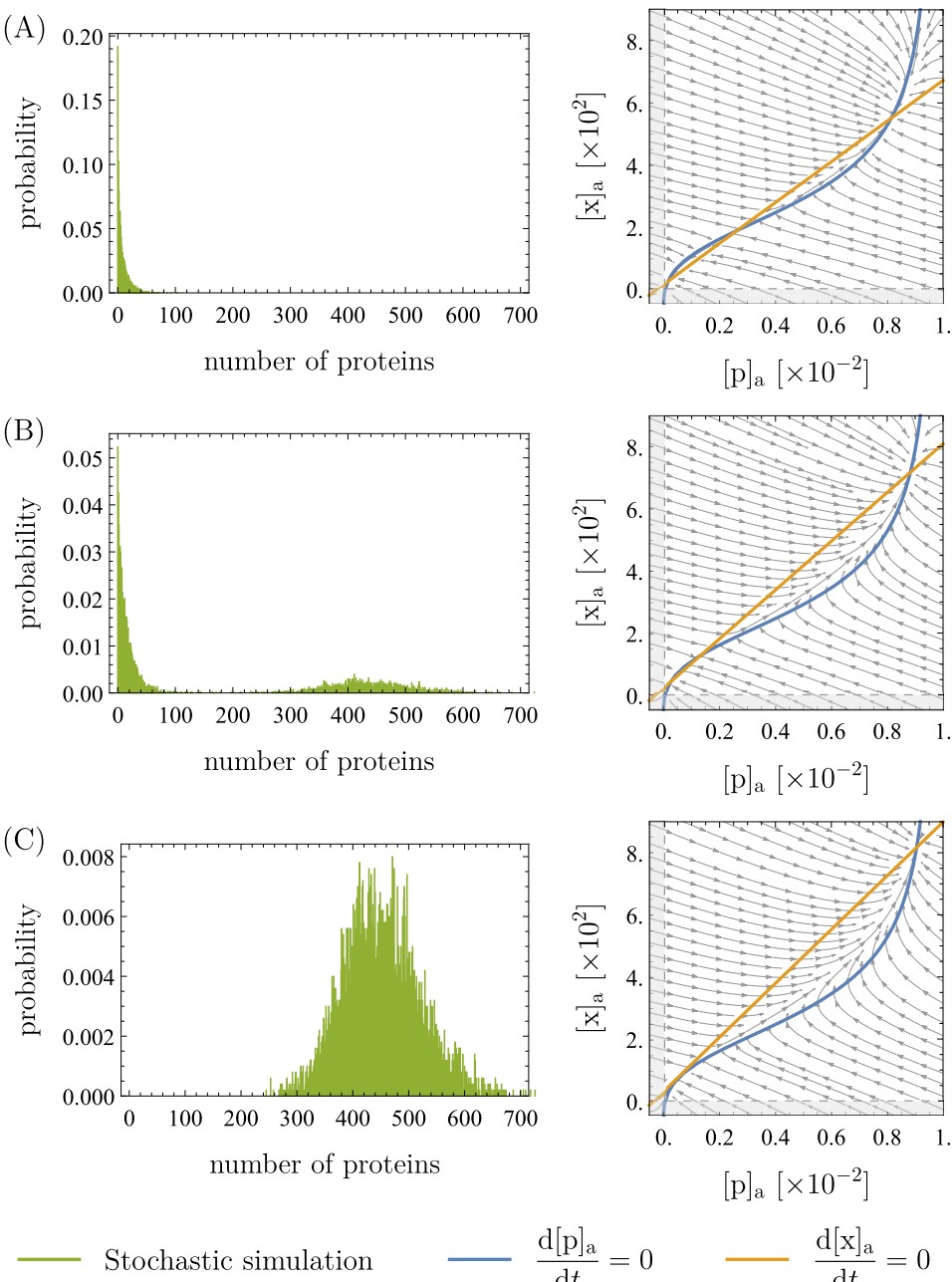

**Fig 9. Distributions from stochastic simulations and the corresponding deterministic phase portraits.** Apart from $[c]_a$, the parameters are the same as in Table 1. The phase portraits were obtained from the deterministic system similarly to those presented in the previous sections. For the distributions, the simulations were run 5000 times for $10^6$ s each (simulated time) and started at a mRNA, protein and intracellular xanthosine count of 0. We show the two cases of unimodality (low expression in (A) and high expression in (C)) as well as the case of bimodality in (B). The values of $[c]_a$ are 12 in (A), 18.5 in (B), and 25 in (C) (recall that $[c]_a := \frac{c}{K_a}$ with $K_a = 5 \cdot 10^{-5}$ M, so $[c]_a$ is dimensionless). The output from the stochastic simulations is in good agreement with the concentrations at the fixed points in the deterministic phase portraits.

cells will actually populate both the high and the low expression state, because they do not show the effective barrier height between the two states. In Fig 9(A), a deterministically bistable scenario is shown where the cells never switched to the high expression state during the run time of our simulations. This result implies that bistability, characterized by two separate *stable* steady-states, largely persists even in the presence of stochastic fluctuations, suggesting that the deterministic picture is remarkably effective. In other words, the fact that bimodality only occurs with some fine tuning of parameters means that the circuit is a strong switch and the deterministic picture is sufficient except in a small region of parameter space. Whether or not the system is actually in that region of parameter space remains a question for future experiments.

To elaborate on the preceeding statements, we found that the two lower fixed points (marked as 1 and 2 in Fig 5) need to be very close like in Fig 9(B) to give bimodality. For lower $[c]_a$, meaning larger difference between the concentrations at the first and second fixed point, almost no switching was observed. Of course, switching is also a matter of the waiting time and stochastic effects: if one waits for long enough, it should eventually occur. However, switching times of more than several hours are not at the center of this investigation and would mean that switching is extremely unlikely. There are two aspects that become relevant in this context that we neglect in our analysis but briefly mention here: transcription and translation bursts lead to higher stochasticity and cell division leads to some discontinuity in the process. The effect of bursts is addressed in S1 Text.

Note that while the deterministic analysis assumes the variables to be continuous, the simulations work with discrete numbers of mRNA, protein, and xanthosine. This per se is no problem, because the deterministic analysis describes the mean values and the simulation fluctuates around this mean. However, if the mRNA number of the third (high) fixed point is so low that stochastic fluctuations are larger than the difference in concentrations between the first and the second fixed point, the system may not be able to resolve the two points anymore. The tolerance to this is surprisingly large, though: In Fig 9, the distance between the first and the third fixed point is around 3 mRNA molecules. While this is around the size of the fluctuations in Fig 9, the latter become as large as 10 mRNA molecules when bursts are included in the simulation (see S1 Text) and yet, the system is able to resolve the fixed points very well.

**Time evolution and adaptation times.** Fig 10A shows the time evolution of the protein concentration for one typical run of the simulation. Again, the simulation was started with a mRNA, protein and intracellular xanthosine count of 0 and was run for a simulated time of $5 \cdot 10^5$ s. In this specific example, the fixed point was reached after roughly $1.5 \cdot 10^5$ s (adaptation time). This time varies: Fig 10B shows the distribution of adaptation times from 1000 runs of the simulation. In both figures, the trajectory obtained from integrating the deterministic system is shown for comparison. It clearly agrees well with the simulation on average.

Comparing this to experimentally expected timescales is difficult, because the adaptation time strongly depends on the extracellular xanthosine concentration. Experiments were always stopped after a few hours, and in this time, the cell population might not reach its steady-state expression distribution. Hence, the distribution could be bimodal when the experiment is stopped but become unimodal after further waiting. That way, extracellular xanthosine concentrations that are too high for deterministic bistability could lead to experimental bimodality if the experiment is stopped too early. In this case, the observed timescales would be shorter, which makes the comparison to our simulations even harder. Thus, we cannot say if it is problematic that the $10^5$ s is larger than what was found in the experiment.

Nevertheless, we do warn the reader that the timescales in the simulations and even more so in the deterministic system should be taken with reservation. Fluctuations in the parameters are not considered here, and neither are cell divisions or the burstiness of transcription and

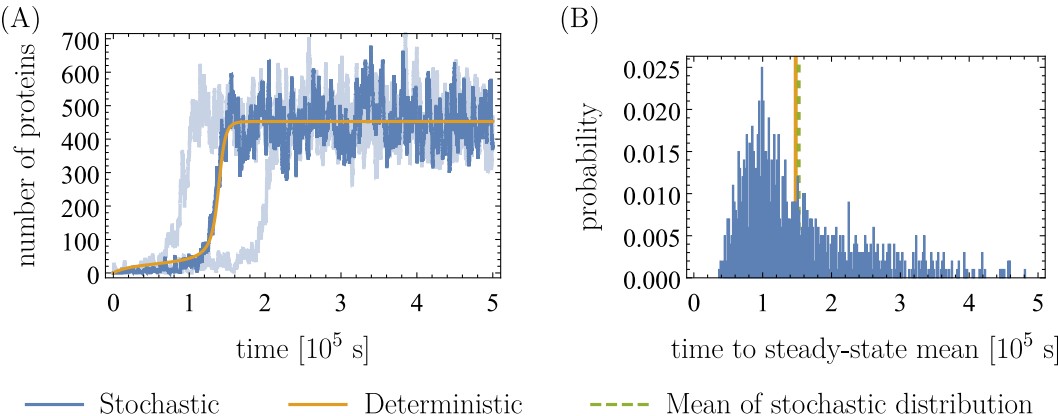

**Fig 10. Stochastic and deterministic time evolution of protein (XapA/XapB) and adaptation time.** (A) Shown in blue is the result from one typical run of the stochastic simulation, and in orange the trajectory obtained from solving the deterministic ODE's. In light blue, two more extreme runs of the simulation are shown for comparison. The simulation was run for $5 \cdot 10^5$ s each and started at an mRNA, protein and intracellular xanthosine count of 0. The parameters that were used are the same as in Table 1, the only exception being the extracellular xanthosine concentration, which was chosen to be $[c]_a = 25$ (recall $[c]_a := \frac{c}{K_a}$, $K_a = 5 \cdot 10^{-5}$ M) just as in Fig 9(C). (B) In blue we plot the waiting time distribution for 1000 runs of the simulation (same conditions as (A)) to reach the steady-state mean. We define this time as the elapsed time when the protein copy number first reaches 90% of its value at the upper fixed point. To better visualize the bulk of the distribution, we excluded from the plot $\sim 10$ runs with adaptation times larger than $6 \cdot 10^5$ s. The green dashed line indicates the mean of the blue distribution. The orange line shows the corresponding deterministic time.

translation. This means that stochasticity may be larger in the real system which should have an influence on the timescales and may shorten the time until the fixed point is reached. However, including transcriptional bursts in the stochastic simulation changes little in the output: the qualitative behavior remains the same but fluctuations around the mean as well as in the adaptation time become larger and bimodality already occurs at lower $[c]_a$ (see S1 Text). Still, the fact that there are no big changes shows that the system is stable to stochastic perturbations and our particular assumptions should not be too significant.

Of course, the system moves to its mean steady-state much faster when the extracellular xanthosine concentration is further away from the bimodal regime (in analogy to a "critical point"). Note that it could well be that in reality, the extracellular xanthosine concentration is high enough to be in the regime where there only is the upper fixed point and thus no bistability or even bimodality. As the system reaches its mean steady-state faster in that regime, the bacteria could adapt more quickly.

**Bifurcation diagram and hysteresis.** In Figs 9 and 10, the simulations were started at initial intracellular concentrations of 0 to investigate what happens if xanthosine is suddenly added to the cell's environment. We can now ask the opposite question: what happens when xanthosine is removed from the extracellular environment? To answer this, the simulation was started with initially fully induced cells, i.e. at the mRNA, protein and intracellular xanthosine counts of the high fixed point in the corresponding phase portrait.

The distributions we obtained can be found in S1 Text. Here, we instead present the results in the form of a bifurcation diagram in Fig 11. In blue, the positions of the deterministic fixed points for the corresponding extracellular xanthosine concentration are shown. As explained before, there is only one fixed point for low and high values of $[c]_a$, but in between, there are three. In yellow and orange, the results from the stochastic simulation are shown, where the mean of each distribution was taken. For the orange points, the simulation was started at $[m]_a = [p]_a = [x]_a = 0$, leading to the distributions in Fig 9. In contrast, the yellow points result from

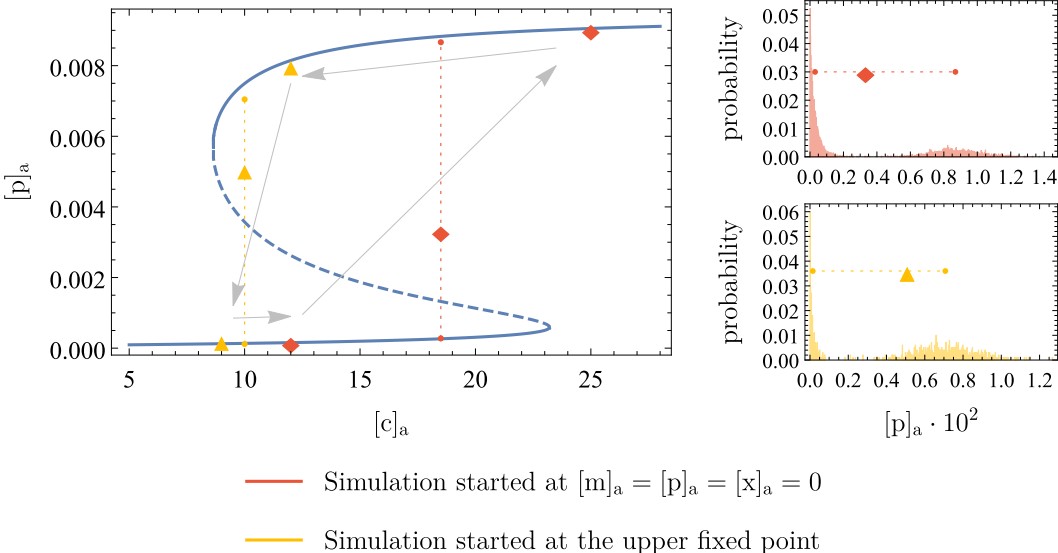

Simulation started at $[m]_a = [p]_a = [x]_a = 0$

Simulation started at the upper fixed point

**Fig 11. Bifurcation diagram showing the hysteresis loop in the stochastic system.** The blue line shows the positions of the deterministic fixed points, while the dashed line indicates the instability of the middle one. The orange and yellow points show the mean of the stochastic simulations when started at zero and at the high fixed point, respectively. The positions of the two peaks in the bimodal distributions are indicated by smaller points, connected by dotted lines. To make this clearer, the bimodal distributions themselves are shown on the right. The arrows illustrate the hysteresis loop in the stochastic simulations.

starting the simulation at the $[m]_a$, $[p]_a$, and $[x]_a$ values of the high fixed point. Note that by taking the mean of each distribution, we hide the feature of bimodality. For that reason, we additionally show the approximate position of the two peaks in the bimodal distribution. We do this by estimating the position of the minimum between the two peaks of the distribution, splitting it into two parts there, and calculating the mean of each of the parts. Also note that it suffices to show the bifurcation diagram in one dimension (here protein concentration) because this uniquely determines the fixed points (recall from Fig 5, the fixed points are unique points in the 3D system).

The figure clearly shows the hysteresis in the system: there exist extracellular xanthosine concentrations where initially uninduced cells remain uninduced and initially induced cells remain induced. Only when the second and the third fixed point are very close can initially induced cells "switch off" to the uninduced state. This behavior is symmetric to the "switching on" in the previous paragraphs. The arrows in Fig 11 indicate the hysteresis loop.

In other words, cells only change their metabolism to xanthosine if enough of the latter is around, but after they have switched, this metabolic state is stable even if the xanthosine concentration decreases to a certain extent. This stability explains what was observed by Novick and Weiner [5] for the *lac* operon: when induced cells were transferred into lower concentrations of lactose, they remained induced, even though uninduced cells could not become induced at these concentrations.

In addition to the hysteresis, Fig 11 also illustrates the astonishingly good agreement of the stochastic simulation and the deterministic results, despite having copy numbers below 10 in some cases. Although one should be cautious about this because of higher stochasticity in the real biological system (addressed above), the result does show that the switch-like feature in the circuit is strong and stable.

## Conclusion

Here, we propose a simple model for genetic circuits containing a membrane transporter whose gene expression is, directly or indirectly, activated by its substrate. We have shown that such a system can be bistable and thus work as a genetic switch which reacts to the extracellular concentration of the relevant metabolite. This switch has very useful biological features. First, coupling of the transporter with, for example, an enzyme which metabolizes the transporter substrate creates a genetic switch that enables short-term adaptation of the cell's metabolism to its environment. Second, the switch is stabilized by hysteresis effects when the extracellular substrate concentration decreases, which explains previous experimental findings. Furthermore, our simulations show that the switch-like behavior is very robust.

However, we have found that no bistability can emerge from the genetic circuit unless at least one component has two binding sites for its activator. Additional binding sites or cooperativity seem to increase the stability of the switch. In addition, simply knowing the experimental switching concentration of xanthosine permits us, for example, to infer the approximate value of the dissociation constant between the transcription factor XapR and the inducer xanthosine. The value we infer is roughly one to two orders of magnitude larger than what has been measured for LacI and IPTG [23], meaning the interaction of XapR and xanthosine is rather weak.

Phase diagrams, showing for which parameters the system is bistable and for which there is only the lower or the upper stable fixed point, could be calculated from arguments made in [29]. However, the simulations showed that in the deterministically bistable regime, which fixed point(s) the system occupies is dependent on initial conditions, which is why we have refrained from showing such phase diagrams. Furthermore, the timescales in the problem could be investigated more thoroughly, for example the dependence of the switching time on $[c]_a$, but such an analysis should probably account for cell divisions and fluctuations in the parameters, which is not straightforward. Lastly, it could be interesting to investigate the magnitude of the fluctuations around the fixed point away from and near the bifurcations in $[c]_a$.

Despite the small copy numbers at the lower fixed point, the stochastic simulations are in excellent agreement with deterministic predictions. All model parameters could be reasonably estimated despite the paucity of experimental knowledge about the model system. The concentrations of mRNA, protein, and xanthosine at the fixed points as well as all qualitative features are as expected from similar systems and the few experiments on the *xap* circuit. These points suggest that the model captures the relevant components of the system correctly and is able to describe its dynamics. The modeling results let us, to some extent, understand how the biological system can achieve its function. By keeping the model as minimal as possible, but still modeling every part explicitly with an appropriate complexity, we can investigate the interesting features while still being able to understand the influence of all parameters and their interplay intuitively.

With the framework given in this text, it should be straightforward to model other promoters, regulatory pathways or enzymes and thereby adapt the model to other genes and metabolites. Examples include *lac*, *ara*, and *xyl*, but we suspect that many if not most metabolic processes involve the adaptation mechanism that we have investigated here, and that much can be understood about them through our model. This apparent success demonstrates once more that even for broadly unknown systems, rigorous physical modeling can potentially offer an efficient way to gain a very thorough understanding of the behavior of the system.

## Materials and methods

### Bacterial strains

Strains used in this work derive from *E. coli* K12 MG1655 with the *lac* operon removed and were constructed similarly to those used in [23, 34]. The Δ*xapABR* knockout was generated using the approach of Datsenko and Wanner [35]. For the fluorescent reporter constructs the *xapAB* promoter region, with one or both XapR binding sites, were extracted from *E. coli* K12 MG1655 by PCR. These were cloned into the YFP-expressing reporter plasmid pZS25 [36], containing kanamycin resistance, and integrated at the *galK* locus using λ Red recombineering [37]. A xapR-mCherry fusion was constructed in a pZS31 plasmid and was integrated into the *ybcN* locus, also using λ Red recombineering. Expression of the xapR-mCherry fusion was regulated by TetR expressed extrachromosomally from a pZS3-PN25 plasmid. Expression of the xapR-mCherry fusion was induced by the addition of 10 ng/mL anhydro-tetracycline. A similar expression system was used in [38] More details can be found in S1 Text.

### Flow cytometry measurements

Experimental measurements in Fig 2 were obtained via flow cytometry and were performed in a similar manner as described in [23]. Briefly, wild-type, Δ*xapABR*, and Δ*xapAB* were grown to saturation in rich LB Miller (BD Medical) medium and were diluted 1:1000 into M9 minimal medium supplemented with 0.5% (w/v) glucose and the appropriate concentration of xanthosine. Cultures were allowed to grow at 37˚C for six to eight hours to an $OD_{600 \text{ nm}}$ of ≈ 0.3. At this point, the cultures were diluted 1:10 into M9 + 0.5% glucose and were immediately analyzed via a MacsQuant VYB Flow Cytometery (MiltenyiBiotec). Measurement files were exported to CSV file formats and analyzed as previously described in [23].

## Supporting information

**S1 Text. The aforementioned further information.** Discussion of simplifications in the model, parameter estimation, elaborations on the results, and the chemical master equation of this circuit. Experimental materials and methods.
(PDF)

## Acknowledgments

We thank Jane Kondev and Jin Wang for interesting discussions, Patrick Lenggenhager for his help with *Mathematica* issues, Nigel Orme for assistance with illustrations, and the anonymous reviewer for many helpful comments. Portions of the experiments reported here were carried out at the Physiology Course at the Marine Biological Laboratory in Woods Hole, MA, operated by the University of Chicago.

## Author Contributions

**Conceptualization:** Kathrin S. Laxhuber, Muir J. Morrison, Rob Phillips.

**Formal analysis:** Kathrin S. Laxhuber.

**Investigation:** Griffin Chure, Charlotte Strandkvist, Kyle L. Naughton.

**Methodology:** Kathrin S. Laxhuber, Muir J. Morrison, Rob Phillips.

**Resources:** Nathan M. Belliveau.

**Writing – original draft:** Kathrin S. Laxhuber.

**Writing – review & editing:** Muir J. Morrison, Griffin Chure, Nathan M. Belliveau, Rob Phillips.

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
