## [Decision Letter · Decision Letter 0]

22 Jan 2020

PONE-D-19-32846

Theoretical investigation of a genetic switch for metabolic adaptation

PLOS ONE

Dear Dr. Phillips,

Thank you for submitting your manuscript to PLOS ONE. After careful consideration, we feel that it has merit but does not fully meet PLOS ONE’s publication criteria as it currently stands. Therefore, we invite you to submit a revised version of the manuscript that addresses the points raised during the review process.

We would appreciate receiving your revised manuscript by Mar 07 2020 11:59PM. To enhance the reproducibility of your results, we recommend that if applicable you deposit your laboratory protocols in protocols.io, where a protocol can be assigned its own identifier (DOI) such that it can be cited independently in the future. For instructions see: http://journals.plos.org/plosone/s/submission-guidelines#loc-laboratory-protocols

We look forward to receiving your revised manuscript.

Kind regards,

L. Michel Espinoza-Fonseca

Academic Editor

PLOS ONE

Journal Requirements:

**Comments to the Author**

1. Is the manuscript technically sound, and do the data support the conclusions?

Reviewer #1: Partly

2. Has the statistical analysis been performed appropriately and rigorously? 

Reviewer #1: Yes

3. Have the authors made all data underlying the findings in their manuscript fully available?

Reviewer #1: No

4. Is the manuscript presented in an intelligible fashion and written in standard English?

Reviewer #1: Yes

5. Review Comments to the Author

Reviewer #1: In their manuscript "Theoretical investigation of a genetic switch for metabolic adaptation", Laxhuber and colleagues present a very nice theoretical analysis of the behavior of the E. coli xapABR system, using both deterministic and stochastic simulations to demonstrate that the system can show meaningful and important bistability driven by the activity of the membrane transporter. In general the work is well done, and it is technically of high quality, but at present I have two substantial concerns that I think must be addressed in order for the paper to live up to its potential.

My first concern is that while the authors present their deterministic simulations first, and draw many of their conclusions primarily from those simulations, it seems to me that the use of such simulations is contra-indicated by the authors' own findings. As both the number of XapR molecules and mRNA molecules are small integers (likely 10 or less), treatment of these species as continuous, deterministic quantities can never be more than a poor approximation. I find this troublesome for a couple of reasons. First, the authors still treat their continuous treatment as an apparently 'good' way to model a genetic switch like that at work at the xapAB promoter; while I find their analysis insightful, in the end I cannot support such a recommendation given the stochasticity intrinsic in the low protein and mRNA copy numbers present here. Second, while the authors do technically recapitulate the bistable behavior of the system in the stochastic version, the bistability seems based on their description to be quite fragile and not very meaningful, since it only occurs here for small distances between the lower and upper fixed points (and seems to take an unreasonably long time to switch between states). I think the authors are likely correct in their assertion that they underestimate the amount of noise in the system due to not including transcriptional and translational bursting, and that the transition rates would likely be faster, but this gives no idea of how *much* faster, and thus it seems premature to simply declare victory and move on.

My recommendation would be that the authors put in some more cautionary words about the applicability of the continuous model, and ideally spend a little more time characterizing the discrete model. in terms of switching times, parameters that give rise to bimodal behavior, effects of incorporating reasonable amounts of transcriptional bursting, etc. Some more direct comparisons of the behavior of the two models in similar ranges of parameters space would also likely be enlightening.

My other major difficulty here is that essentially all of the work is premised on reference 22, which is 'unpublished data.' This is the only experimental evidence given for the bistability of xapAB expression, justification of many of the model parameters, experimentally expected switching timescales, etc. It is impossible to properly interpret the present work, either for the purposes of review or from the viewpoint of an interested reader, without availability of those data, which either ought to be their own independent publication or ought to be included with this one.

6. PLOS authors have the option to publish the peer review history of their article (what does this mean?). If published, this will include your full peer review and any attached files.

Reviewer #1: No

---

## [Author Response · Author response to Decision Letter 0]

10 Mar 2020

We have uploaded our letter in the "Attach Files" section, as a PDF titled "Response to Reviewers".

---

## [Decision Letter · Decision Letter 1]

27 Mar 2020

PONE-D-19-32846R1

Theoretical investigation of a genetic switch for metabolic adaptation

PLOS ONE

Dear Dr. Phillips,

Thank you for submitting your manuscript to PLOS ONE. After careful consideration, we believe the manuscript was substantially improved, and the reviewer is impressed with the changes made to the manuscript. However, the reviewer has raised a few (minor) clarifying questions that need to be addressed before the manuscript can be accepted for publication. Therefore, we invite you to submit a revised version of the manuscript that addresses the points raised during the review process.

We would appreciate receiving your revised manuscript by May 11 2020 11:59PM. To enhance the reproducibility of your results, we recommend that if applicable you deposit your laboratory protocols in protocols.io, where a protocol can be assigned its own identifier (DOI) such that it can be cited independently in the future. For instructions see: http://journals.plos.org/plosone/s/submission-guidelines#loc-laboratory-protocols

We look forward to receiving your revised manuscript.

Kind regards,

L. Michel Espinoza-Fonseca

Academic Editor

PLOS ONE

Reviewers' comments:

Reviewer's Responses to Questions

**Comments to the Author**

1. If the authors have adequately addressed your comments raised in a previous round of review and you feel that this manuscript is now acceptable for publication, you may indicate that here to bypass the “Comments to the Author” section, enter your conflict of interest statement in the “Confidential to Editor” section, and submit your "Accept" recommendation.

Reviewer #1: (No Response)

2. Is the manuscript technically sound, and do the data support the conclusions?

Reviewer #1: Yes

3. Has the statistical analysis been performed appropriately and rigorously? 

Reviewer #1: Yes

4. Have the authors made all data underlying the findings in their manuscript fully available?

Reviewer #1: Yes

5. Is the manuscript presented in an intelligible fashion and written in standard English?

Reviewer #1: Yes

6. Review Comments to the Author

Reviewer #1: In general the authors have done an admirable job addressing my comments. At this point, I find myself confused on one issue, which I hope the authors will address (unless I'm just being truly dense). In the right side of Figure 9, a series of phase portraits are displayed. I assume that these must have been from the deterministic system, since the authors make no reference to performing a suitable scan of conditions to obtain such diagrams for the stochastic system; I assume that these in fact arise from the deterministic system. However, the figure legend does not address this point at all, and the reader is left with the impression that those phase portraits arise from the stochastic system. The issue is not helped by the fact that the coloring and axis scaling for the right side of Figure 9 does not follow the same conventions used in earlier diagrams; for example, the right panel of Fig 9A looks a lot like Fig. 5B, but the colors and axis scales are different.

I would also like to caution the authors (and editorial staff) that while the "Revised Manuscript with Track Changes" document was fully readable, the figure numbering of the main text figures seemed to be off in the submission system.

As a final note, I would like to urge the authors to make the source code for their simulations publicly accessible, which would greatly help the community in adapting similar models for other systems.

7. PLOS authors have the option to publish the peer review history of their article (what does this mean?). If published, this will include your full peer review and any attached files.

Reviewer #1: No

---

## [Author Response · Author response to Decision Letter 1]

1 Apr 2020

We uploaded our response letter in the "Attach Files" section under the name "Response to Reviewers".

---

## [Editor Report · Decision Letter 2]

2 Apr 2020

Theoretical investigation of a genetic switch for metabolic adaptation

PONE-D-19-32846R2

Dear Dr. Phillips,

We are pleased to inform you that your manuscript has been judged scientifically suitable for publication and will be formally accepted for publication once it complies with all outstanding technical requirements.

With kind regards,

L. Michel Espinoza-Fonseca

Academic Editor

PLOS ONE

---

## [Editor Report · Acceptance letter]

16 Apr 2020

PONE-D-19-32846R2 

Theoretical investigation of a genetic switch for metabolic adaptation 

Dear Dr. Phillips:

I am pleased to inform you that your manuscript has been deemed suitable for publication in PLOS ONE. Congratulations! Your manuscript is now with our production department. 

With kind regards,

on behalf of

Dr. L. Michel Espinoza-Fonseca 

Academic Editor

PLOS ONE